# Prototype-Guided Hyperbolic Multi-Scale Learning and Density-Ratio Analysis for Subtype-Specific Nuclei Discovery in WSIs

## Abstract

We present a self-supervised framework that discovers subtype-specific nuclei directly from whole-slide images (WSIs) by learning a unified, multi-scale representation in a single hyperbolic space. Lymphoma progression manifests as coordinated morphological changes across scales-from individual nuclei to tissue architecture-yet most existing pipelines encode these scales with separate models and rely on tissue-level supervision, which obscures cell-level drivers of subtype identity. Our approach replaces this separation with inclusion-aware self-supervision: local crops (nucleus patches) and their containing global crops (tissue patches) are jointly embedded in a Poincaré ball, whose hyperbolic geometry naturally accommodates hierarchical structure. A cross-scale alignment objective pulls each nucleus toward the representation of its parent tissue region, enabling a single encoder to capture fine-to-global morphology without cell-level labels. Tissue patches sampled from lesions typically carry features sufficient for subtype discrimination and form clusters reflecting both lesional status and subtype. By contrast, single-nucleus patches are often weakly informative in isolation and inter-subtype differences at the tissue level largely arise from composition ratios of common nuclear phenotypes. This approach yields nucleus-level exemplars and spatial patterns that are consistent with tissue-level subtype structure while preserving interpretability at cellular resolution.

## 1 Introduction

### 1.1 Objective

Pathologists diagnosing malignant lymphoma first examine tissue morphology in low-magnification images to identify lesional regions within the target whole-slide image (WSI) and to estimate the lymphoma subtype. They then observe the morphology of individual cells in the identified regions at higher magnification to reinforce the confidence of the subtype estimation. In the subtype addressed in this study, known as Follicular Lymphoma (FL), for example, cells called centroblasts increase in number as the cancer progresses (Swerdlow et al., 2017). The nuclei of centroblasts are larger in area compared to those of other cell types. The proportion of such subtype-specific cells improves the accuracy of subtype diagnosis, and explicitly highlighting these nuclei in an automated system enhances interpretability (Koga et al., 2024).

We present a self-supervised framework that discovers subtype-specific nuclei directly from whole-slide images (WSIs) by learning a unified, multi-scale representation in a single hyperbolic space. Lymphoma progression manifests as coordinated morphological changes across scales-from individual nuclei to tissue architecture-yet prevailing pipelines encode these scales with separate models and rely on tissue-level supervision, which obscures cell-level drivers of subtype identity. Our approach replaces this separation with inclusion-aware self-supervision: local crops (nucleus patches) and their containing global crops (tissue patches) are jointly embedded in a Poincaré ball, whose hyperbolic geometry naturally accommodates hierarchical structure. A cross-scale alignment objective pulls each nucleus toward the representation of its parent tissue region, enabling a single encoder to capture fine-to-global morphology without cell-level labels.

In this study, we propose a self-supervised approach that learns a unified representation of the multi-scale hierarchy observed in malignant lymphoma, from single-nucleus images to large tissue patches, within a single hyperbolic (Poincaré) space. As disease progresses, nuclear morphology and tissue architecture evolve in a coordinated manner, and image representations should therefore reflect cross-scale dependencies. To this end, we perform representation learning driven by inclusion relations between images (Fig. 1, left): when one patch spatially contains another, their embeddings are encouraged to be proximal in feature space.

Most existing multi-scale pipelines encode cell-level and tissue-level images with separate encoders, and impose hierarchy by feeding cell-level features into a tissue-level network (e.g., U-Net (Ronneberger et al., 2015), HIPT (Chen et al., 2022)). Cancer subtype classification is typically performed from tissue-level images (e.g., via multiple instance learning), whereas classifying subtypes from single-nucleus patches is generally difficult. Moreover, not all regions in a whole-slide image (WSI) exhibit subtype-specific alterations, and even within lesional areas many nuclei lack subtype-specific morphology. Consequently, supervised hierarchical learning tends to make subtype discrimination achievable at the tissue level but limits the extent to which cell-level embeddings capture subtype variation.

We address these issues by embedding multi-scale images-from $56 \times 56$ nucleus crops to $224 \times 224/112 \times 112/56 \times 56$ tissue crops-into a single hyperbolic feature space using a shared encoder. The learned geometry captures both cross-scale appearance similarity and the inclusion-based hierarchy: when one patch contains another, the encoder maps them to nearby points. Representing nested, tree-like structures in a Euclidean space poses capacity limitations, whereas hyperbolic spaces are known to embed such hierarchies with low distortion. During training, representations of global (tissue) images are directly influenced by the included local (nucleus) images, and vice versa. In our setting, the smallest local unit is a single-nucleus image; if certain nuclear morphologies are subtype-specific, their embeddings will lie near the embeddings of tissue images of that subtype.

Since each nuclear patch inherits the slide-level label of its parent tissue, we estimate subtype-specific probability densities over nuclear embeddings in the learned space and compute, at each location in the space, a density ratio-the density of a given subtype relative to that of the remaining subtypes. Nuclear images whose embeddings lie where this ratio markedly exceeds one are extracted as subtype-specific nuclei, yielding interpretable nucleus exemplars. This operationalizes the discovery of subtype-specific nuclei.

## 1.2 RELATED WORKS

Two representative approaches have been proposed for modeling hierarchical representations in histopathological images.

### 1.2.1 HIERARCHICAL TRANSFORMER-BASED REPRESENTATION

In this framework, each hierarchical level from cells to tissue regions is represented by a Transformer (Chen et al., 2022; Grisi et al., 2023; Guo et al., 2023; Buzzard et al., 2024; Tang et al., 2025). Images at each level and location are encoded via class tokens. Higher layers cover wider fields of view and generate new class tokens by aggregating those from the lower level. The lowest layer typically represents cell-level patches and is often trained with self-supervision, whereas the top layer is trained with subtype labels.

### 1.2.2 GRAPH NEURAL NETWORK (GNN)-BASED REPRESENTATION

Since pathology fundamentally regards nuclei as the unit of analysis, GNN-based methods (Li et al., 2018; Pati et al., 2020; Adnan et al., 2020; Brussee et al., 2025; Mirabadi et al., 2025) are widely explored. After segmenting nuclei, each nucleus is represented as a node and connected to neighboring nuclei to form a graph. A GNN then aggregates topological and spatial relations among nuclei, producing tissue-level representations at higher layers.

Both methods share the principle of aggregating microscopic (nuclear-level) features to obtain macroscopic (tissue-level) representations. However, this design makes it difficult for tissue-level appearances to influence nuclear-level embeddings. By contrast, our proposed method leverages inclusion relations, embedding nuclear and tissue patches in parallel within a shared space, such that

representations at different scales are mutually constrained. Notably, CNN- and Transformer-based approaches do not explicitly encode individual nuclei even at the cellular scale, while GNNs treat each nucleus as a node but their aggregation does not directly capture tissue-level appearances. Our method explicitly constructs nuclear patches through segmentation and links them to weakly magnified tissue appearances via inclusion, thereby bridging micro- and macro-level representations.

### 1.3 PRELIMINARIES

#### 1.3.1 HYPERBOLIC SPACE

The proposed method adopts Poincaré ball (Ganea et al., 2018; Gromov, 1987; Nickel & Kiela, 2017; Tifrea et al., 2018) as the feature space. Unlike Euclidean space, the volume of Poincaré ball grows exponentially with the distance from the origin. The Poincaré ball allows embedding a greater number of points as they move farther from the origin, so it is well-suited for representing hierarchical data (see Fig. 2 (Left)).

The Riemannian metric of the $d$-dimensional Poincaré ball $\mathbb{B}_\kappa^d := \{z \in \mathbb{R}^d | \kappa\|z\|^2 < 1\}$, where $\kappa > 0$ is the (negative) curvature, is given as follows:

$$g_\mathbb{B}(z) = \frac{4}{\left(1 - \kappa\|z\|^2\right)^2} g_\mathbb{E} \tag{1}$$

where $z \in \mathbb{B}_\kappa^d$ is the position vector, and $g_\mathbb{E}$ is the standard metric of Euclidean space (Durrant & Leontidis, 2023; Ge et al., 2023).

In the Poincaré ball model, the radius $r$ and the curvature $\kappa$ satisfy the following relation:

$$\kappa = \frac{1}{r^2} \tag{2}$$

In this study, the curvature and radius of the Poincaré ball are fixed at $\kappa = 1$, $r = 1$. As $\|z\|$ approaches 1, the coefficient of $g_\mathbb{E}$ increases, causing movements to become larger as $z$ moves away from the center. The geodesic distance on the Poincaré ball between two points $z_1, z_2 \in \mathbb{B}_\kappa^d$ can be computed as follows:

$$d_\mathbb{B}(z_1, z_2) = \frac{2}{\sqrt{\kappa}} \operatorname{artanh}\left(\sqrt{\kappa}\| - z_1 \oplus_\kappa z_2\|\right) \tag{3}$$

The exponential map of the Poincaré ball, which projects Euclidean embeddings onto the Poincaré ball is given by:

$$\exp_v^\kappa(x) := v \oplus_\kappa \left(\tanh\left(\frac{\sqrt{\kappa}\|x\|}{1 - \kappa\|v\|^2}\right)\frac{x}{\sqrt{\kappa}\|x\|}\right) \tag{4}$$

where $v \in \mathbb{B}_\kappa^d$ is set the origin of the Poincaré ball, and $x \in T_z\mathbb{B}_\kappa^d \approx \mathbb{R}^d$. This exponential map converts a Euclidean latent space to a hyperbolic one. $\oplus_\kappa$ denotes Möbius addition, a differentiable operation defined as follows:

$$x \oplus_\kappa y = \frac{(1 + 2\kappa\langle x, y\rangle + \kappa\|y\|^2)x + (1 - \kappa\|x\|^2)y}{1 + 2\kappa\langle x, y\rangle + \kappa^2\|x\|^2\|y\|^2} \tag{5}$$

In the proposed method, an embedding $z \in \mathbb{B}_\kappa^d$ on the Poincaré ball is assigned to each prototype $c$ placed on the boundary $\partial\mathbb{B}_\kappa^d$ of the Poincaré ball. The boundary $\partial\mathbb{B}_\kappa^d$ corresponds to the sphere $\mathbb{S}^{d-1}$ and is not included in the Poincaré ball. To evaluate the assignment of $z$ to a prototype $c$, we employ the Busemann function (Bridson & Haefliger, 2013; Busemann, 2012; Ghadimi Atigh et al., 2021):

$$b_c(z) = \log\left(\frac{\|c - z\|^2}{1 - \|z\|^2}\right) \tag{6}$$

## 2 METHOD

### 2.1 PROPOSED METHOD

The proposed method adopts the loss function used in Hyperbolic Masked Siamese Networks (HMSN) (Durrant & Leontidis, 2023). HMSN uses prototypes (Assran et al., 2022; Caron et al.,

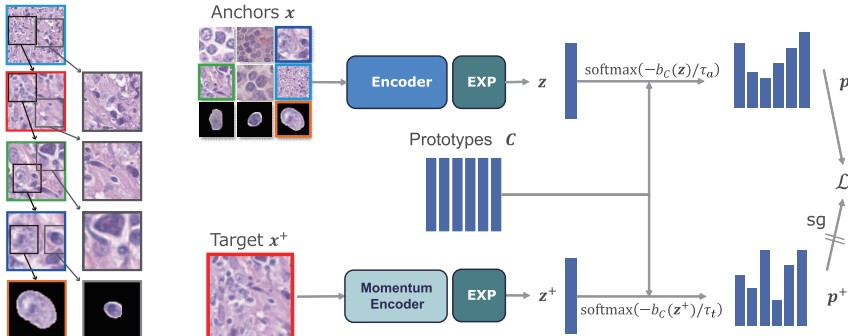

Figure 1: (Left) Inclusion relationships between images, from a tissue image down to a cell nucleus image. (Right) Self-supervised learning using hyperbolic embeddings and prototypes. Image pairs with inclusion relationships, as shown on the left, are selected as anchor and target.

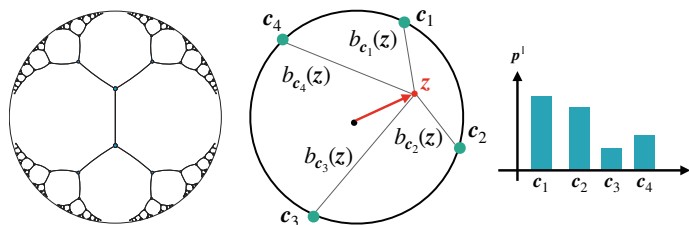

Figure 2: (Left) Embedding tree structures into 2-dimensional poincaré ball (Nickel & Kiela, 2017). All branches within the Poincaré ball are of equal length. (Right) Position representation using prototypes placed on the boundary of the Poincaré ball.

2020) to structure representations in hyperbolic space. In the proposed method, the prototypes placed on the boundary of the Poincaré ball, guide the embeddings of images based on the inclusion relationships between the global and local morphology.

The proposed method extracts tissue images, $x_i$, from WSIs. For each $x_i$, one target view and $M$ anchor views with inclusion relationships are generated. The target view is denoted as $x_i^+$ and anchor views as $x_{i,m}$ where $m = 1, 2, \cdots, M$. In contrast to HMSN, where masking and conventional image augmentation are used to generate the anchor and targets from an image, our method constructs anchor-target pairs from images of different spatial scales based on their inclusion relationships (see Fig. 1 (Right)). This enables cross-scale learning of hierarchical features. Here, the scale of an image refers to the crop size when extracting it from $x_i$. Each is mapped to the Poincaré ball using Eq. (4) to obtain $z_i^+ \in \mathbb{B}_\kappa^d$ and $z_{i,m} \in \mathbb{B}_\kappa^d$ respectively. $K$ learnable prototypes $c_k \in \mathbb{R}^d$ are placed on the Poincaré ball boundary $\partial\mathbb{B}_\kappa^d$, and the assignment $b_{c_k}(z)$ from $z$ to each $c_k$ are computed using the Busemann function shown in Eq. (6).

Let the set of $K$ computed assignments be denoted by a $K$-dimensional vector, $b_c(z) = [b_{c_1}(z), \ldots, b_{c_K}(z)]^\top$. Each feature vector $z$ is represented by using the prototype assignment vector $p \in \Delta_K$ computed as follows:

$$p := \text{softmax}\left(\frac{-b_c(z)}{\tau_*}\right) \quad (7)$$

In Fig. 2 (Right), an example of a 2-dimensional Poincaré ball with four prototypes is shown. The representation $z$ embedded in the Poincaré ball possesses a 4-dimensional prototype assignment vector.

Following HMSN, we use different temperature parameters for the target and anchor views in the soft assignment step. Let $\tau_t$ denote the temperature for the target view and $\tau_a$ denote the temperature for the anchor view. We set $\tau_t = 0.0625$ and $\tau_a = 0.25$ in all experiments using pathology images. Among the images used for encoding, cell nucleus images have the smallest scale. As for

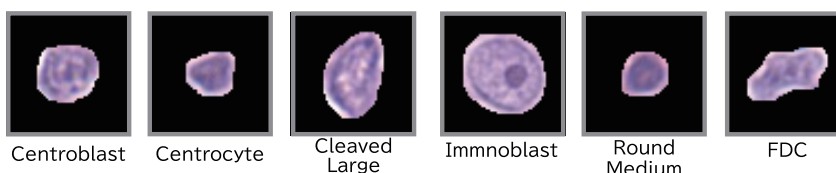

Centroblast Centrocyte Cleaved Large Immnoblast Round Medium FDC

Figure 3: Examples of six types of cell nucleus images. Different morphologies and textures can be observed for each type.

cell nucleus images, they are generated by segmenting cell nuclei, cropping each nucleus centered at its centroid, and setting the pixel values outside the cell nucleus to zero aiming to isolate the morphological features of individual nuclei.

The distance between $\boldsymbol{p}_i^+$ obtained from the target view and $\boldsymbol{p}_{i,m}$ obtained from the anchor view is evaluated using the cross-entropy $\frac{1}{MB}\sum_{i=1}^{B}\sum_{m=1}^{M} H(\boldsymbol{p}_i^+, \boldsymbol{p}_{i,m})$. To ensure that all prototypes are utilized evenly, a regularization term is introduced to maximize the entropy $H(\bar{\boldsymbol{p}})$ of the average the prototype assignment vector across all anchor views, where $\bar{\boldsymbol{p}} = \frac{1}{MB}\sum_{i=1}^{B}\sum_{m=1}^{M}\boldsymbol{p}_{i,m}$ (Assran et al., 2021). Additionally, to encourage anchor views to be assigned to specific prototypes, the term $\frac{1}{MB}\sum_{i=1}^{B}\sum_{m=1}^{M} H(\boldsymbol{p}_{i,m})$ is minimized. The encoder and prototypes are trained to minimize the following objective function:

$$\mathcal{L} = \frac{1}{MB}\sum_{i=1}^{B}\sum_{m=1}^{M} H(\boldsymbol{p}_i^+, \boldsymbol{p}_{i,m}) - \lambda H(\bar{\boldsymbol{p}}) + \beta\frac{1}{MB}\sum_{i=1}^{B}\sum_{m=1}^{M} H(\boldsymbol{p}_{i,m}) \tag{8}$$

where $\lambda$ and $\beta$ are hyperparameters.

## 2.2 DATASET AND IMPLEMENTATION

### 2.2.1 DATASET

From a malignant lymphoma database, approximately 5,000 WSIs with subtype diagnoses were selected, and from these, around 150,000 tissue images were obtained by randomly extracting $4096 \times 4096$ tissue regions from histological sections. The selection of WSIs was not restricted by subtype, and the extraction of tissue images did not prioritize specific histological features, such as follicles, which could contribute to disease classification. It should be noted that the number of subtypes included in the training data was more than 70, and many of the sampled images originated from regions within WSIs that are less likely to contribute to subtype classification. The dataset included a significant number of tissue images from subtypes other than Reactive, Follicular Lymphoma (FL), and Diffuse Large B-Cell Lymphoma (DLBCL), which were later used for validation.

To generate target and anchor views, tissue images were cropped from random positions within the $4096 \times 4096$ tissue images. Three different crop sizes were used: $224 \times 224$, $112 \times 112$, and $56 \times 56$. During training, disease subtype labels were not used and were referenced only for evaluation. To obtain individual cell nucleus images from the tissue images, HoVer-Net (Graham et al., 2019) was applied for cell nucleus segmentation. Regardless of nucleus size, each nucleus image was extracted as a fixed-size $56 \times 56$ patch, with the surrounding region outside the nucleus set to zero. In total, approximately 2.9 billion cell nucleus images were generated and used.

Cell nuclei within follicles play a crucial role in diagnosing Reactive and FL cases (Swerdlow et al., 2017). Therefore, cell type labels were assigned to the nuclei inside follicles in WSIs of Reactive and FL cases by an expert, categorizing them into seven classes: Round-Medium, Centrocyte, Centroblast, Immunoblast, Cleaved-Large, Follicular Dendritic Cell (FDC), and Others. Since DLBCL does not contain follicles, nuclei from randomly selected tissue images were labeled by the expert pathologists. These annotations were not used during self-supervised learning but were utilized for analyzing embeddings in the Poincaré ball.The number of labeled cell nucleus images for each cell type is as follows. Round-Medium has 561 images, Centrocyte has 683 images, Centroblast has 239 images, Immunoblast has 29 images, Cleaved-Large has 123 images, and FDC has 93 images (As an example, randomly selected cell nucleus images from each cell type are shown in Fig. 3).

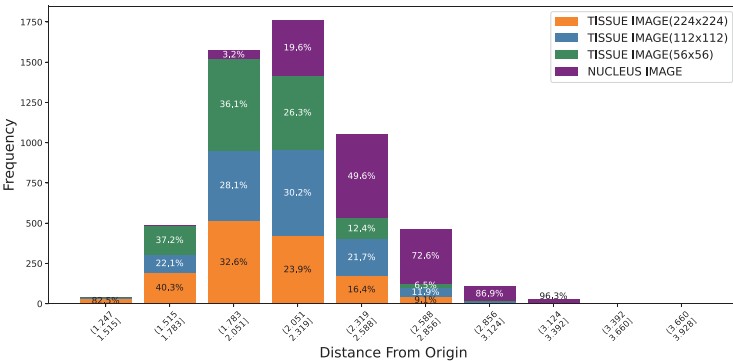

Figure 4: Distance of tissue and cell nucleus image representations from the origin: Nucleus images, represented with hard-masked backgrounds, tend to be located near the boundary. Tissue images reside closer to the origin.

### 2.2.2 IMPLEMENTATION

In this study, the encoder is based on ResNet34 (He et al., 2016), with the final linear layer replaced by a multilayer perceptron, and an additional exponential mapping shown in Eq. (4) applied. The dimension of the Poincaré ball is set to $d = 64$. The loss function coefficients in Eq. (8) are set to $\lambda = 10$ and $\beta = 0.25$. The target temperature in Eq. (7) is set to $\tau_t = 0.0625$, and the anchor temperature is set to $\tau_a = 0.25$. The number of prototypes is set to $K = 512$. The model was trained for $500$ epochs with a batch size of $1024$. We used RiemannianAdam (Becigneul & Ganea, 2019) for optimizer. In training, the first 50 epochs are used as warm-up epochs, during which the learning rate is linearly increased to 1e-4. After that, a cosine scheduler is used to decrease the learning rate to 1e-5.

### 2.2.3 LIMITATIONS

In our current setup, nuclear images are created by extracting a $56 \times 56$ patch centered on each segmented nucleus, with all pixels outside the nuclear region set to zero. This hard-masked background, while helpful for isolating nuclear morphology, introduces an artificial boundary that could bias representation learning. Moreover, the nucleus segmentation is performed by HoVer-Net (Graham et al., 2019), and inaccuracies in segmentation particularly in dense or overlapping regions could introduce noise or bias into the downstream representation space. Future work should consider more nuanced strategies, such as soft masking, confidence-aware sampling, or incorporating cytoplasmic features where feasible.

## 3 EXPERIMENTS AND RESULTS

### 3.1 DISTRIBUTION OF PATHOLOGICAL IMAGES ON THE POINCARÉ BALL

We analyze the distribution of tissue and nuclear image representations on the Poincaré ball after learning. Representations are expected to form a continuous hierarchy by distance from the origin (Nickel & Kiela, 2018; Yang et al., 2023). Fig. 4 shows the discretized distribution of nuclear and other tissue image representations by their distance from the origin. Purple graph shows nucleus images, green graph shows tissue images (cropped $56 \times 56$), blue graph shows tissue images (cropped $112 \times 112$), and orange graph shows tissue images (cropped $224 \times 224$). In the figure, the horizontal axis represents the geodesic distance from the origin, while the vertical axis indicates the frequency at each geodesic distance. The numerical values shown are percentages. The representation of the cell nucleus images, shown in purple, increases in proportion as the distance from the origin increases. It can be confirmed that the representation of the tissue images is distributed closer to the origin compared to the representation of the cell nucleus images. From these results, we believe we have represented a hierarchical structure, where cell nucleus images are positioned farther from the origin and tissue images are positioned closer to the origin.

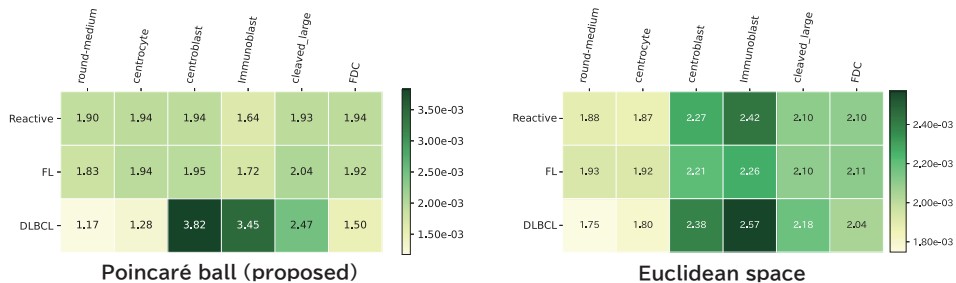

Figure 5: Heatmap showing the assignment degrees between subtypes (columns) and annotated nucleus types (rows). Higher values indicate that the corresponding tissue and nucleus images tend to be assigned to similar prototypes. Results obtained by the Proposed method (Left) and by using the Euclidian space for the latent space (Right). The left table indicates subtype-specific patterns in prototype usage, particularly for two cell-types, Centroblasts and Immunoblasts in one subtype, DLBCL.

### 3.2 AFFINITY ANALYSIS TISSUE AND CELL NUCLEUS IMAGES

To investigate how the learned prototype assignments reflect subtype-specific patterns, we compute the prototype assignment vector $\boldsymbol{p}$ between each disease subtype and each cell nucleus type. Since the embedded images are labeled with disease subtypes, we calculate the mean assignment vector for each subtype. Let the average prototype assignment vectors for Reactive, FL, and DLBCL be denoted as $\boldsymbol{p}_{\text{Reactive}}, \boldsymbol{p}_{\text{FL}}, \boldsymbol{p}_{\text{DLBCL}} \in \Delta K$. If tissue images of a given subtype are consistently assigned to certain prototypes, the corresponding components of $\bar{\boldsymbol{p}}_{\text{subtype}}$ will take larger values.

Some nuclei were also labeled by nucleus type, and we similarly compute the mean prototype assignment vector $\bar{\boldsymbol{p}}_{\text{cell}}$ for each nucleus type. The alignment between a disease subtype and a nucleus type is evaluated as the inner product of their mean vectors, $\bar{\boldsymbol{p}}_{\text{subtype}} \cdot \bar{\boldsymbol{p}}_{\text{cell}}$, which quantifies the similarity between the two distributions. A higher value indicates that nucleus and tissue images of the given types tend to be assigned to the same prototypes.

The results are shown in left panel in Fig. 5. Note that during self-supervised training, many additional subtypes beyond Reactive, FL, and DLBCL were included, and subtype labels were not referenced. Cancer progression typically follows the order Reactive → FL → DLBCL.

As shown in left panel in Fig. 5, Centroblasts, Immunoblasts, and Cleaved Large cells increase with disease progression, while Centrocytes decrease-a trend most pronounced in DLBCL. Moreover, Round Medium and Centrocytes are strongly associated with Reactive and FL but show weaker associations with DLBCL. Importantly, the observed increase in Centroblasts and decrease in Centrocytes with progression is consistent with WHO grading criteria for FL (Swerdlow et al., 2017).

For comparison, we repeated the same training procedure using a Euclidean space. The parameters learned with our proposed method were retained, but representation learning was performed in Euclidean space, and prototype assignments were computed using Eq. (9), which replaces the Busemann function with cosine similarity.

$$\boldsymbol{p} := \text{softmax}\left( \frac{\langle \boldsymbol{z}, \boldsymbol{c} \rangle}{\tau_*} \right) \tag{9}$$

The Euclidean results, shown in right panel in Fig. 5, indicate that relationships between certain cell types (e.g., Immunoblasts, Centroblasts) and the DLBCL subtype are less distinct. This suggests that in Euclidean space, prototypes for DLBCL may not be well separated from those of Reactive or FL, limiting the ability to form subtype-specific nuclear clusters around corresponding tissue embeddings. By contrast, embedding in a Poincaré ball provides sufficient capacity to preserve subtype-specific organization.

Finally, to validate our method in a simplified setting, we applied the same inclusion-based learning framework to the MNIST dataset. The results are provided in Appendix A.2.

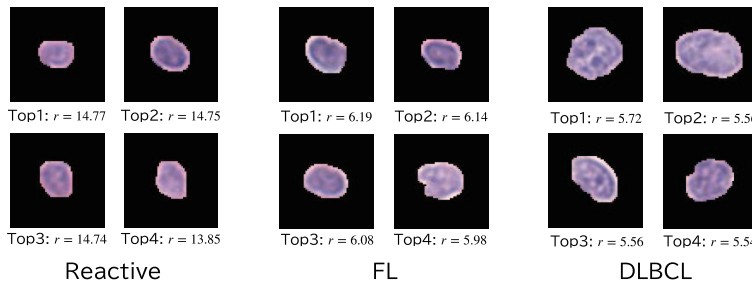

Figure 6: Top four nuclear images with the highest density ratios discovered by *the proposed method* for each subtype. From left to right: Reactive, FL, and DLBCL. The numerical values below the images indicate the density ratio for the disease subtype predicted by that image. As cancer progresses, nuclei of increasing size are selected as subtype-specific exemplars.

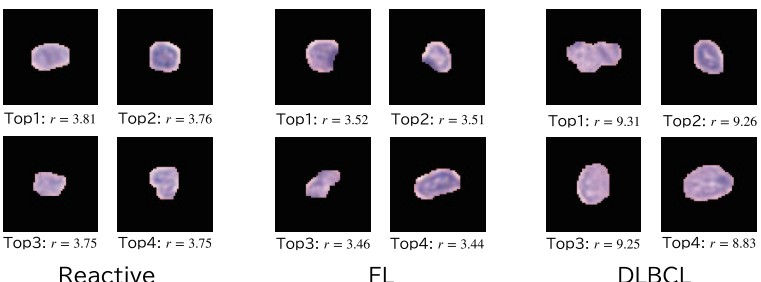

Figure 7: Top four nuclear images with the highest density ratios obtained from training in *Euclidean space*. Compared with Fig. 6, the trend of increasing nuclear size with disease progression is not successfully captured.

### 3.3 DISCOVERY OF SUBTYPE-SPECIFIC CELL NUCLEI USING DENSITY RATIO FUNCTION

In the previous experiment, we evaluated the validity of the learned representations by comparing them with established pathological knowledge regarding relationships between subtypes and cell types. For example, our method embedded nuclear images of centroblasts and tissue images of the DLBCL subtype into the same prototype cluster. Next, we examine whether the proposed method can directly identify subtype-specific nuclei-for instance, whether centroblasts can be discovered as nuclei specific to DLBCL.

After training with the proposed method, we focus solely on the embeddings of nuclear images. Each nuclear image is assigned one of the three subtype labels inherited from its source WSI. Let the probability density distribution of embeddings for subtype A be denoted as $q_A(z)$. If nuclei specific to subtype A exist, they should be isolated from the distributions of the other two subtypes B and C. To evaluate this, we compute the density ratio function:

$$r_A(z) = \frac{q_A(z)}{q_{B+C}(z)} \tag{10}$$

where higher values of $r_A$ indicate stronger subtype specificity. For density-ratio estimation we employ Kullback-Leibler Importance Estimation Procedure (KLIEP) (Sugiyama et al., 2007), which directly estimates the ratio without separately estimating each density. KLIEP also allows the kernel scale parameter for ratio estimation to be adaptively determined by likelihood cross-validation (LCV) from the data (Sugiyama et al., 2007). In this experiment, each nuclear image is thus evaluated with three density ratios: $r_{\text{Reactive}}$, $r_{\text{FL}}$, and $r_{\text{DLBCL}}$.

After obtaining the density ratio functions, $r_{\text{Reactive}}$, $r_{\text{FL}}$, and $r_{\text{DLBCL}}$, we evaluated the values of the three density ratio functions for each of 190,000 cell nucleus images not utilized in the obtaining of the density ratio functions.

Next, all nuclear images were sorted in descending order according to their density ratios for each subtype. Fig. 6 presents examples of nuclei with high density ratios for each subtype. It can be

Table 1: Proportion of each nucleus type among the top $\alpha\%$ of nuclear images with the highest density ratios for subtype DLBCL. A higher density ratio of DLBCL is associated with a higher proportion of centroblasts, which is consistent with the WHO criteria (Swerdlow et al., 2017).

| Cell-type $\alpha$ | Round Medium | Centrocyte | Centroblast | Reject |
|---|---|---|---|---|
| 0.005 % | 0.00 % | 0.00 % | **100.00 %** | 0.00 % |
| 0.05 % | 0.00 % | 19.05 % | **66.67 %** | 14.29 % |
| 0.5 % | 21.30 % | 15.74 % | **51.85 %** | 11.11 % |
| 5 % | 27.17 % | 18.85 % | **45.10 %** | 8.87 % |

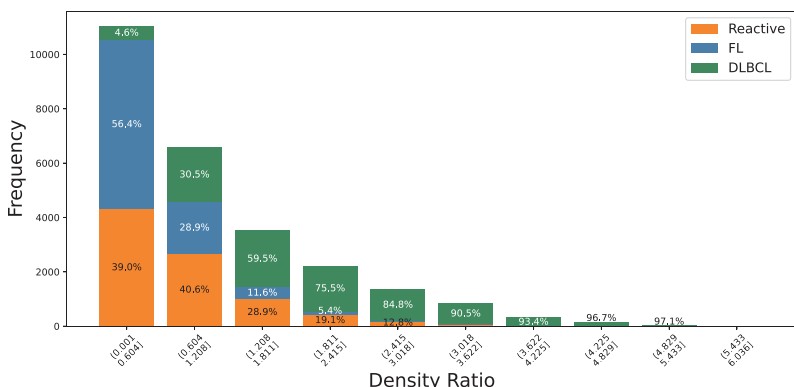

Figure 8: Histogram of density ratios for each subtype based on approximately 8700 detected Centroblast nuclei from all images, regardless of subtype. All nuclei with DLBCL density ratios greater than 0.11 were classified as Centroblasts. Thus, nuclei selected as DLBCL-specific were identified as Centroblasts.

observed that nuclei of progressively larger size are identified in the order of Reactive, FL, and DLBCL. The results of examining the top 10 cell nucleus images for each disease subtype in this experiment are presented in Appendix A.3. For comparison, we also performed the same representation learning in Euclidean space and estimated density ratios in the same manner. The results are shown in Fig. 7. Unlike Fig. 6, these results make it difficult to confirm the pathological finding that nuclear enlargement accompanies lymphoma progression.

Next, after sorting by density ratio, we performed nucleus-type classification. For constructing the classifier, we used (Cao et al., 2022), which outputs a Reject label when the confidence of type estimation is low. Table 1 summarizes the classification results for the top $\alpha\%$ of nuclear images (ranked by density ratio) of DLBCL. By selecting nuclei with high density ratios, our method successfully identified centroblasts. For reference, a single WSI contains approximately 1.5 to 2.5 million nuclei. Although $\alpha = 0.005\%$ may seem small, lesional regions within a WSI are spatially limited, and within these regions, several dozen *specific* nuclei are included. More detailed pathological analysis of these results remains a subject for future work.

Fig. 8 shows the distribution of density ratios for nuclei identified as Centroblasts, with respect to Reactive, FL, and DLBCL. As illustrated, nuclei labeled as Centroblasts exhibit higher density ratios for DLBCL, a finding consistent with established pathological knowledge.

## 4 CONCLUSION

We proposed a method to represent malignant lymphoma pathology images in a single Poincaré ball using self-supervised learning, covering images from cell nuclei to tissue scale. By leveraging inclusion relationships across scales, our method embeds tissue and contained nucleus images in close proximity. Future work will extend this analysis to a broader range of lymphoma subtypes and statistical significance tests.

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

# A APPENDIX

## A.1 LLM USAGE

We used a large language model for the purpose of refining grammar and syntax in the final manuscript. All conceptual and analytical content was independently written by the authors.

## A.2 VERIFICATION USING THE MNIST DATASET

This section describes the results of training conducted on the MNIST dataset (Deng, 2012) using images created by introducing inclusion relationships, in order to verify the expressions obtainable through the proposed method. The MNIST dataset, which has an inclusion relationship, provides images showing a single digit alongside images showing four adjacent digits or the same digit. Here, an adjacent digit refers to either the digit $a+1$ or the digit $a-1$ for a given digit $a$ ($a = 0, 1, \cdots, 8, 9$). In this case, digits 0 and 9 have only one adjacent digit each. To ensure every digit has exactly two adjacent digits, we define a cyclic relationship among adjacent digits, such that digit 0 and digit 9 are considered adjacent to each other. The inclusion relationship defined within the dataset is shown in Fig. 9. An image containing a single digit has an inclusion relationship with images containing multiple adjacent digits and with images containing multiple instances of the same digit. All images have a size of $56 \times 56$.

Unlike the experiments with malignant lymphoma pathology images, we set the dimension of the Poincaré ball to $d = 3$ and trained ResNet50 (He et al., 2016) with $K = 64$ prototypes. Additionally, a temperature of $\tau_t = 0.0375$ for the target image and $\tau_a = 0.15$ for the anchor image. Training was performed for 2000 epochs with a batch size of 200. Other settings are also the same.

We report the results of affinity analysis using prototype assignment vector based on prototype $c$ for post-learning representations. This analysis examines two types of image representations for the digits 0 and 1. The first type includes two images: one showing a single digit 0 and one showing a single digit 1. The second type of image includes five different images: an image showing four

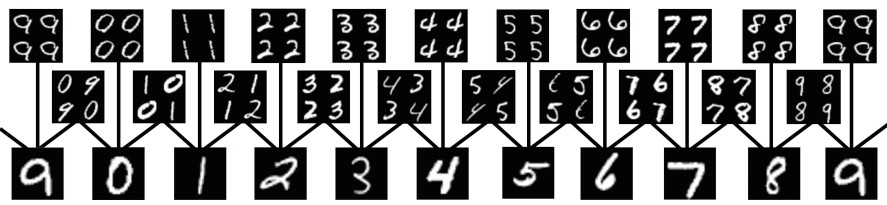

Figure 9: Visualization of inclusion relationships set in the dataset via graphs.

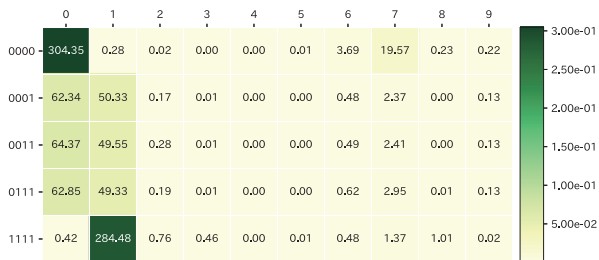

Figure 10: Heatmap showing the assignment degrees between image showing four numbers (columns) and image showing a single digit (rows). Higher values indicate that the corresponding images tend to be assigned to similar prototypes. It can be confirmed that images containing a single digit and images containing four digits within them show a particularly strong correlation.

zeros, an image showing three zeros and one one, an image showing two zeros and two ones, an image showing one zero and three ones, and an image showing four ones. Each image is encoded by the pre-trained encoder, and the average of the prototype assignment vectors $p$ obtained from the distance to the prototypes is calculated. By calculating the dot products between these average prototype assignment vectors, we verified how the representations are positioned relative to the prototypes. Based on the defined inclusion relationships, we expect that images displaying four digits will exhibit varying relationships with images displaying a single digit, depending on the specific digits contained within them. The calculation results are shown as a heatmap in Fig. 10.

From the results shown in Fig. 10, it can be confirmed that for the digits 0 and 1, images containing one digit 0 or one digit 1, and images containing a total of four digits, show a stronger relationship compared to images containing one digit of another type. Furthermore, the images containing four identical digits and those containing two different digits each show changes in the related images. It is evident that images containing four identical digits exhibit particularly strong relationships only with the images appearing within that specific image. Here, the prototype assignment vector is calculated based on the distance from the prototype, and thus reflects the position within the representation space. Therefore, images containing four identical digits and images containing two different types of digits are thought to be positioned differently within the image, corresponding to the digits depicted. Furthermore, since the placement is positioned near the image containing the single digit within the inclusion relationship, we believe the proposed method achieves placement corresponding to the digit appearing within the image for the MNIST dataset possessing inclusion relationships.

### A.3 SUBTYPE PREDICTION BASED ON DENSITY RATIOS

This chapter presents the top 10 density ratios for each disease subtype and the predicted cell nucleus images used for disease subtype classification based on cell nucleus images. Additionally, the density ratios $r_{\text{Reactive}}$, $r_{\text{FL}}$, and $r_{\text{DLBCL}}$ calculated for each cell nucleus image are presented in a table below the nuclear images.

#### A.3.1 CELL NUCLEI IMAGES PRESUMED TO BE REACTIVE

Fig. 11 shows ten cell nuclei images predicted to be of the Reactive.

In cell nuclei predicted to be of the Reactive, $r_{\text{Reactive}}$ yielded the highest value, followed by DLBCL as the next most significant subtype. Furthermore, it can be confirmed that many of the cell nucleus types inferred from the cell nucleus images are also classified as Round Medium or Centrocyte. As $r_{\text{Reactive}}$ decreases, no change can be observed where the density ratio of other disease subtypes increases.

#### A.3.2 CELL NUCLEI IMAGES PRESUMED TO BE FL

Fig. 12 shows ten cell nuclei images predicted to be of the FL.

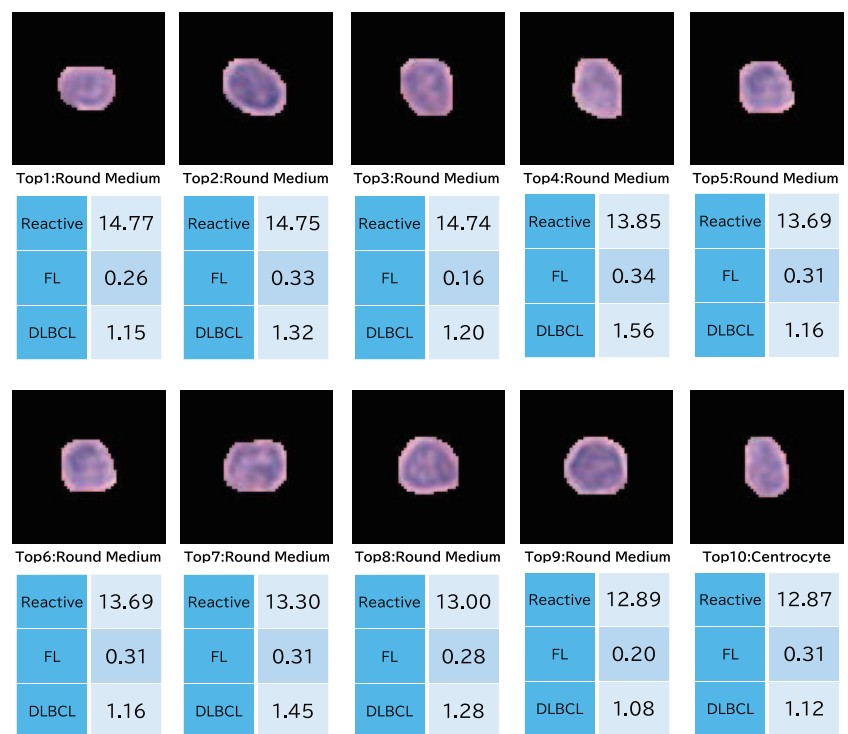

Figure 11: Ten cell nucleus images predicted as Reactive. The table beneath each image shows the values obtained when calculating three density ratios for that image. These images confirm that the density ratio for Reactive is significantly higher compared to other disease subtypes.

In cell nuclei predicted to be FL, $r_{\text{Reactive}}$ and $r_{\text{DLBCL}}$ are approximately 1 or less, suggesting low similarity to other disease subtypes and high similarity to FL. The inferred cell nucleus types are Round Medium and Centrocyte, similar to the results for Reactive. Similarly, no increase in the density ratio of other disease subtypes has been observed in response to a decrease in the density ratio of FL.

### A.3.3 CELL NUCLEI IMAGES PRESUMED TO BE DLBCL

Fig. 13 shows ten cell nuclei images predicted to be of the DLBCL.

The cell nuclei predicted to be DLBCL are larger in shape compared to nuclei predicted to be other disease subtypes. Regarding cell nucleus types, many nuclei were predicted to be centroblasts. Furthermore, as with other disease subtypes, it is not observed that as the density ratio of DLBCL decreases, the density ratios of other disease subtypes increase.

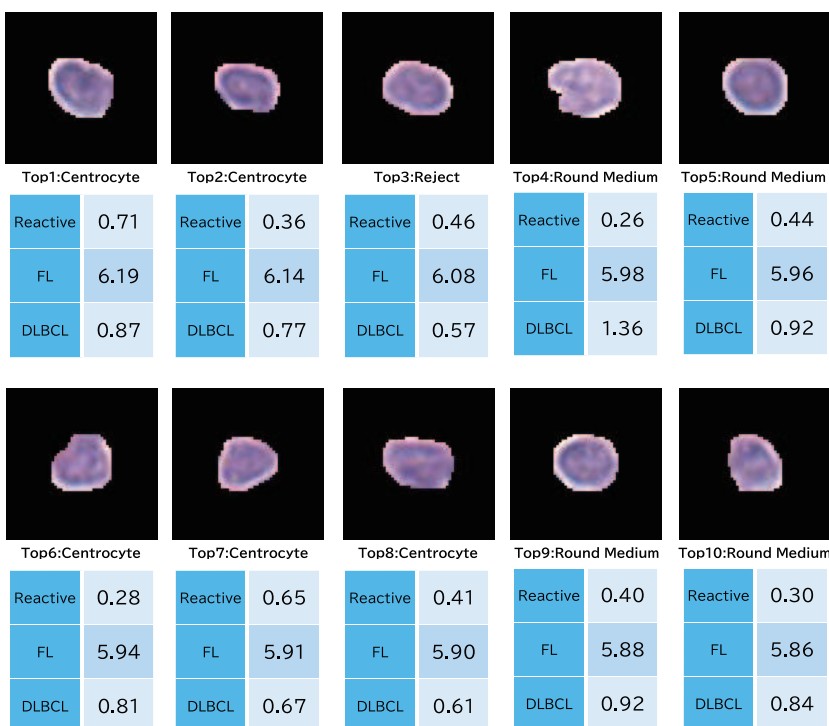

Figure 12: Ten cell nucleus images predicted as FL. The cell nuclei predicted to be FL showed a density ratio of approximately 1 or less relative to other disease subtypes, suggesting a high degree of FL-like characteristics.

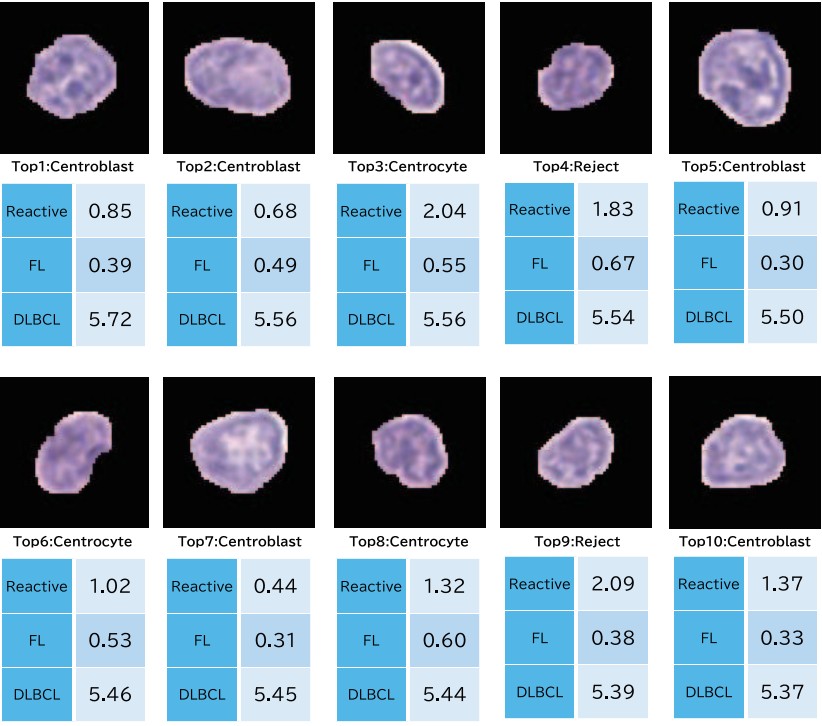

Figure 13: Ten cell nucleus images predicted as DLBCL. Both DLBCL and predicted cell nuclei images show larger shapes compared to other disease subtypes. Furthermore, centroblasts are the predominant cell type.

