# OpenReview forum: "Prototype‑Guided Hyperbolic Multi‑Scale Learning and Density‑Ratio Analysis for Subtype‑Specific Nuclei Discovery in WSIs"
_ICLR.cc/2026/Conference — ICLR 2026 Conference Withdrawn Submission_

### Official Review · Reviewer_f9uV · 2025-10-22

**Soundness:** 1
**Presentation:** 3
**Contribution:** 2
**Rating:** 4
**Confidence:** 2

**Summary:**

This paper proposes a self-supervised learning framework for discovering subtype-specific nuclei from Whole-Slide Images (WSIs). The core idea is to learn a unified multi-scale representation by embedding both nuclear patches and their corresponding tissue patches into the same Poincaré ball, a model of hyperbolic space. The self-supervised signal derives from the containment relationships in these spaces, which pulls embedded patches closer to their host container patches in the embedding space, following a training approach similar to Hyperbolic Masked Siamese Networks (HMSN). After training a shared encoder, the model leverages WSI-level subtype labels to estimate probability density ratios on nuclear embeddings, identifying nuclei with high density ratios as representative instances of that subtype. The authors present qualitative results demonstrating that this method can identify clinically meaningful cellular features, such as the enrichment of larger centroblasts in Diffuse Large B-Cell Lymphoma (DLBCL). Furthermore, the paper demonstrates the superiority of hyperbolic geometric spaces over Euclidean spaces in handling such hierarchical tasks.

**Strengths:**

1. The paper uses spatial containment relationships, which inherently encode hierarchical structures and domain knowledge, as supervisory signals in self-supervised learning to embed multi-scale feature information into a unified representation space. This constitutes a core idea that is both novel and elegant.
2. Some evidence demonstrate that hyperbolic space is more appropriate than Euclidean space for learning representations that respect spatial containment structures in pathological images.
3. Discovering interpretable, subtype-specific cellular biomarkers directly from whole-slide images (WSIs) without cell-level annotations is a problem of great clinical significance and research value. The two-stage approach proposed in this paper—first performing unsupervised representation learning, then conducting density ratio–based discovery—offers a reasonable methodology for addressing this problem.

**Weaknesses:**

1. Insufficient experimental validation. The paper lacks comparative experiments with any state-of-the-art self-supervised learning methods in representation learning for computational pathology. Furthermore, no comparisons are conducted on downstream tasks such as whole slide image classification. This makes it impossible to evaluate the quality of the learned features within the context of this domain.
2. The paper includes many hyperparameters, such as $\lambda$, $\beta$, and the temperature  in softmax, but lacks a sensitivity analysis of these hyperparameters.
3. The evaluation of the "discovered" nuclei is primarily qualitative or correlational. While the visual examples are compelling, there is a lack of rigorous quantitative evaluation.

**Questions:**

1. It would be helpful to provide comparative experimental results against other state-of-the-art self-supervised methods, as well as comparative results on downstream tasks, to validate the effectiveness of the proposed method.
2. How would the recall and precision of cell subtype classification be if directly analyzing cell subtypes based on probability density ratios?

---

### Official Review · Reviewer_B6Yr · 2025-10-25

**Soundness:** 3
**Presentation:** 2
**Contribution:** 3
**Rating:** 4
**Confidence:** 4

**Summary:**

The paper proposes a self-supervised framework that learns unified multi-scale representations within a single hyperbolic space, aiming to automatically discover subtype-specific nuclei in WSIs. Although the idea of hyperbolic multi-scale representation is conceptually interesting, the paper lacks quantitative experiments, ablation analyses, and direct comparisons with existing WSI representation methods, which makes the effectiveness of the proposed approach unclear and questionable.

**Strengths:**

1. The paper explores an unsupervised approach to learning unified representations from the nuclear to the tissue level within a single embedding space. This direction is exploratory and meaningful in pathological representation learning.

2. The paper introduces the novel idea of using hyperbolic space to achieve multi-scale consistent representations, modeling the inclusion relationships between cells and tissues as geometric hierarchies. This design leverages hyperbolic geometry to encode hierarchical biological structures more effectively than conventional Euclidean embedding.

3. The hyperbolic embedding visualizations and density-ratio analyses demonstrate the biological interpretability, suggesting that the learned representations capture meaningful subtyp morphological variations.

**Weaknesses:**

1. Although the proposed method is conceptually novel, the experimental scope is limited. Most importantly, the paper lacks any quantitative comparison with existing methods or the baseline model, which makes it difficult to assess the effectiveness of the proposed framework.

2. As a pre-training framework, the experiments mainly focus on qualitative analysis rather than downstream quantitative evaluation. However, the absence of clear performance metrics on downstream applications limits its practical impact. The validation is restricted to a single lymphoma dataset with only three subtypes, without demonstrating generalization to other cancers or tissue types.

3. The loss function is relatively complex with several hyperparameters, yet the paper does not provide ablation studies to verify the contribution of each component. The effects of  $\lambda$, $\beta$, and other hyperparameters  are not analyzed, leaving the robustness and sensitivity of the model unclear.

4. While the dataset appears to be an important part of the contribution, its construction process are not clearly described. It remains ambiguous whether the data were newly collected, or curated from multiple public sources, which raises concerns about reproducibility and data transparency.

**Questions:**

The authors are expected to include sufficient quantitative metrics, comparative experiments, and ablation studies in the rebuttal to substantiate their claims. Hence, despite the novelty of the idea, the technical validation remains far from sufficient for ICLR acceptance.

---

### Official Review · Reviewer_zXkr · 2025-10-30

**Soundness:** 3
**Presentation:** 3
**Contribution:** 2
**Rating:** 2
**Confidence:** 4

**Summary:**

The paper describes a self-supervised framework for discovery of nuclei subtypes in histopathology whole slide images. Authors ideate an interesting conceptual framework to integrate different scales, from nuclear level to tissue level, based on hyperbolic geometry, specifically Poincare' Ball. They assess the proposed framework by targeting  for diagnosis on challenging neoplastic conditions, namely Follicular Lymphoma.

**Strengths:**

1. Manuscript quality: Paper well written and organized; it is very pleasant to read, the schematic diagrams are polished.
2. Novelty: According to my knowledge, the conceptual definition of Hyperbolic Multi Scale Learning is not adequately explored yet, and it appears very interesting. Moreover, I did not find evidence of application of this strategy in computational histopathology.

**Weaknesses:**

1. Design choices and architecture details: authors mention that 56x56 crops are used for detect nucleis (but according to my experience in significant number of cases, this size is not enough to contain the entire nucleus). What does it happen when nuclei are cropped? The proper shape cannot be distinguished properly, and there is high risk of bias also at the aggregation level. Authors should consider adaptive size for that. Not clear in line 208 what is $\tau^*$ (what are the temperature parameters?) How did you get to the taxonomy represented in Fig.3?
2. Positioning of manuscript: while the proposed methodology appears interesting, the assessment provided position it in a field that appears more of interest for the medical community (like MICCAI). Authors should provide more solid rationale about how general is the proposed framework, and how it can be applied to histopathology in general, and not only for a specific use case.
3. Related work: despite the novelty of the framework proposed, I think that a discussion of current trends in self supervision for histopathology is missing, like the works published in Mahmood Lab:
Multimodal Prototyping for cancer survival prediction
Andrew H Song, Richard J Chen, Guillaume Jaume, Anurag Jayant Vaidya, Alexander Baras, Faisal Mahmood
ICML 2024
Morphological prototyping for unsupervised slide representation learning in computational pathology
Andrew H Song, Richard J Chen, Tong Ding, Drew FK Williamson, Guillaume Jaume, Faisal Mahmood
CVPR 2024, or authors can have a look in the survey
Zhang, Y., Gao, Z., He, K., Li, C., & Mao, R. (2025). From patches to WSIs: A systematic review of deep Multiple Instance Learning in computational pathology. Information Fusion, 103027.
Ma, Y., An, B., Shen, A., Yuan, M., Duan, M., & Wang, M. (2025). Flow-MIL: Constructing Highly-expressive Latent Feature Space For Whole Slide Image Classification Using Normalizing Flow. In Proceedings of the IEEE/CVF International Conference on Computer Vision (pp. 23561-23570).

4. Limited assessment: a discussion of outcomes is missing, as well as a proper description of the assessment process. Moreover, since the paper is proposing a self-supervised method for analysis of histopathology images, and various methods have been proposed recently, like the cited Chen et al..   or Song et al., CVPR 2024. Authors should benchmark their method againts other self supervision baseline strategies, on established benchmarks.

**Questions:**

1. Further details about architecture choices, and an analysis on how the framework works at varying of hyperparameters is needed.
2. A revised discussion of the literature about self supervised frameworks for histopathology is needed.
3. Benchmarking on established set of WSIs and against other self supervision strategies is needed.

**Details Of Ethics Concerns:**

No concerns

---

### Official Review · Reviewer_ZzRq · 2025-10-30

**Soundness:** 2
**Presentation:** 1
**Contribution:** 3
**Rating:** 2
**Confidence:** 4

**Summary:**

This paper introduces a self-supervised learning method that integrates inherent hierarchical nature of WSIs by mapping cell-level and tissue-level features in a Poincare ball. Adopting a cross-scale alignment objective, the authors can reveal which cells serve as representative examples for tissue-level predictions, ensuring the interpretability of the proposal.

**Strengths:**

* Using Poincare ball to represent inherent hierarchy in the WSIs is interesting
* Detailed analysis of method in the experiment section.

**Weaknesses:**

* What is the rationale behind the specific scaling parameter value (line 215)?
* Confusing writing. The authors should rearrange and clarify the motivations, problem definitions and the experimental settings.
* Extend literature reviews. Although the authors mentioned prior work on learning multi-scale features, it is not true that there is no prior solution trying to capture this relationship using some type of alignment losses[1,3]. The authors should expand the literature review in this direction for comprehension.
* Unclear benchmark settings. The authors used “a malignant lymphoma database” (line 246) for method evaluation without indicating exactly what it is. Lacking public information about this dataset limits the reproducibility and validation of this work.
* Strong hyperparameter dependence. This method requires manual hyperparameter tuning including loss coefficient, scale temperature, crop size, etc. making the results not trustworthy. If the magnification of the WSIs changes, all these hyperparameters need tuning again to match the results reported in the paper.
* One of advantages of self-supervised methods is to provide general knowledge for deep learning framework for better downstream task tuning. The authors should extend the experiments to show that the proposed framework can learn robust features from a large number of cells-tissue pairs to improve downstream analysis.
* Baselines comparison. There are a lot of self-supervised learning methods [2,3] used to build foundation models in histological images [4,5,6], the authors should compare their proposal with existing solutions both intuitively and empirically to support their claims. The analysis can show the effectiveness of the proposed method but fail to convince us that the proposal is better than current solutions.

[1] Nguyen et al., A Semi-Supervised Learning Framework with Cross-Magnification Attention for Glioma Mitosis Classification, ISBI’25

[2] Oquab et al., DINOv2: Learning Robust Visual Features without Supervision, arxiv’23

[3] Xu et al. A whole-slide foundation model for digital pathology from real-world data, Nature 2024.

[4] Chen et al. Towards a General-Purpose Foundation Model for Computational Pathology, Nature 2024

[5] Lu et al., Visual Language Pretrained Multiple Instance Zero-Shot Transfer for Histopathology Images, CVPR’23

[6] Ma et al., A generalizable pathology foundation model using a unified knowledge distillation pretraining framework, Nature Biomedical Engineering 2025

**Questions:**

See Weaknesses

**Details Of Ethics Concerns:**

The dataset used in this work requires acceptance from some providers, relating to human subjects. However, the authors did not describe the dataset sufficiently.

---

### Note · Authors · 2025-12-01

I have read and agree with the venue's withdrawal policy on behalf of myself and my co-authors.